# CONTEXTUAL KNOWLEDGE DISTILLATION FOR TRANSFORMER COMPRESSION

## ABSTRACT

A computationally expensive and memory intensive neural network lies behind the recent success of language representation learning. Knowledge distillation, a major technique for deploying such a vast language model in resource-scarce environments, transfers the knowledge on individual word representations learned without restrictions. In this paper, inspired by the recent observations that language representations are relatively positioned and have more semantic knowledge as a whole, we present a new knowledge distillation strategy for language representation learning that transfers the *contextual* knowledge via two types of relationships across representations: *Word Relation* and *Layer Transforming Relation*. We validate the effectiveness of our method on challenging benchmarks of language understanding tasks. The code will be released.

## 1 INTRODUCTION

Since the Transformer, a simple architecture based on attention mechanism, succeeded in machine translation tasks, Transformer-based models have become a new state of the arts that takes over more complex structures based on recurrent or convolution networks on various language tasks, e.g., language understanding and question answering, etc. (Devlin et al., 2018; Lan et al., 2019; Liu et al., 2019a; Raffel et al., 2019; Yang et al., 2019) However, in exchange for high performance, these models suffer from a major drawback: tremendous computational and memory costs. In particular, it is not possible to deploy such large models on platforms with limited resources such as mobile and wearable devices, and it is an urgent research topic with impact to keep up with the performance of the latest models from a *small-size* network.

As the main method for this purpose, Knowledge Distillation (KD) transfers knowledge from the large and well-performing network (teacher) to a smaller network (student). Very recently, there have been some efforts that distill Transformer-based models into compact networks (Sanh et al., 2019; Turc et al., 2019; Sun et al., 2019; 2020; Jiao et al., 2019). However, they all build on the idea that each word representation is independent, ignoring relationships between words that could be more informative than individual representations.

In this paper, we pay attention to the fact that word representations from language models are very *structured* and capture certain types of semantic and syntactic relationships. - Word2Vec (Mikolov et al., 2013) and Glove (Pennington et al., 2014) demonstrate that trained embedding of words contains the linguistic patterns as linear relationships between word vectors. Recently, Reif et al. (2019) found out that the distance between words contains the information of the dependency parse tree. Many other studies also suggested the evidence that contextual word representations (Belinkov et al., 2017; Tenney et al., 2019a;b) and attention matrices (Vig, 2019; Clark et al., 2019) contain important relations between words. Intuitively, although each word representation has respective knowledge, the set of representations of words as a whole is more semantically meaningful, since words in the embedding space are positioned relatively by learning.

Inspired by these observations, we propose a novel distillation objective, termed Contextual Knowledge Distillation (CKD), for language tasks that utilizes the statistics of relationships between word representations. In this paper, we define two types of contextual knowledge: *Word Relation (WR)* and *Layer Transforming Relation (LTR)*. Specifically, WR is proposed to capture the knowledge of *relationships between word representations* and LTR defines *how each word representation changes* as it passes through the network layers. Moreover, unlike some previous approaches with constraints

for distillation, the proposed objective is more robust to architecture changes as it does not add any structural constraints for teacher or student.

There are two distillation techniques to compress a large pre-trained language model into a compact network. Several previous works (Sanh et al., 2019; Jiao et al., 2019; Sun et al., 2020) compress a large pre-trained language model into a small language model on the pre-training stage which requires high computation costs and times. On the other hand, some works (Turc et al., 2019; Sun et al., 2019) present the task-specific distillation that transfers the knowledge to a well initialized small network to improve the performance of each task. In this paper, we focus on the task-specific distillation that has the advantage of being directly applied on top of pre-trained small BERT models (Turc et al., 2019) without conducting a time-consuming pre-training process.

We validate our method on the Stanford Question Answer Dataset (SQuAD) and General Language Understanding Evaluation (GLUE) benchmark. We first demonstrate the effectiveness of our method outperforming the current state-of-the-art distillation methods. We also show that our CKD performs effectively on a variety of network architectures including recently proposed MobileBERT (Sun et al., 2020), a new type of thin architecture of BERT, trained with task-agnostic distillation.

Our contribution is threefold:

- Inspired by the recent observations that word representations from neural networks are structured, we propose a novel knowledge distillation strategy, Contextual Knowledge Distillation (CKD), that transfers the relationships across word representations.
- We present two types of complementary contextual knowledge: horizontal Word Relation across representations in a single layer and vertical Layer Transforming Relation across representations for a single word.
- We validate CKD on the standard language understanding benchmark datasets and show that CKD consistently outperforms the state-of-the-art distillation methods for BERT on various model sizes.

## 2 RELATED WORK

**Knowledge distillation** Since recently popular deep neural networks are computation- and memory-heavy by design, there has been a long line of research on transferring knowledge for the purpose of compression. Hinton et al. (2015) first proposed a teacher-student framework with an objective that minimizes the KL divergence between teacher and student class probabilities. In this framework, several follow-up works proposed various objectives to distill the well-designed knowledge such as attention map of image (Zagoruyko & Komodakis, 2016), similarity (Tung & Mori, 2019) or the relation (Park et al., 2019; Liu et al., 2019b) between the image features.

In the field of natural language processing (NLP), knowledge distillation has been actively studied (Kim & Rush, 2016; Hu et al., 2018; Yang et al., 2020). In particular, after the emergence of large language models based on pre-training such as BERT (Devlin et al., 2018; Liu et al., 2019a; Yang et al., 2019; Raffel et al., 2019), many studies have recently emerged that attempt various knowledge distillation in the pre-training process and/or fine-tuning for downstream tasks in order to reduce the burden of handling large models. Specifically, Tang et al. (2019); Chia et al. (2019) proposed to distill the BERT to train the simple recurrent and convolution networks. Sanh et al. (2019); Turc et al. (2019) proposed to use the teacher's predictive distribution to train the smaller BERT, and Wang et al. (2020) propose the structure-level distillation that transfer the predictive distribution of sequence-level for the multi-lingual sequence labeling tasks. Sun et al. (2019) proposed a method to transfer individual representation of words in the BERT. In addition to matching the hidden state, Jiao et al. (2019) and Sun et al. (2020) also utilize the attention matrices derived from the Transformer. Several works (Goyal et al., 2020; Liu et al., 2020; Hou et al., 2020) improve the performance of other compression methods such as sparsification and quantization by integrating the knowledge distillation objectives. Different from previous knowledge distillation methods that transfer respective knowledge of word representations, we design the objective to distill the contextual knowledge of them, which can be combined with existing distillation methods.

**Contextual knowledge of word representations** Understanding and utilizing the relationships across words is one of the key ingredients in language modeling. Word embedding (Mikolov et al.,

2013; Pennington et al., 2014) that captures the context of a word in a document, has been traditionally used. Unlike the traditional methods of giving fixed embedding for each word, the contextual embedding methods (Devlin et al., 2018; Peters et al., 2018) that assign different embeddings according to the context with surrounding words have become a new standard in recent years showing high performance. Xia & Zong (2010) improved the performance of the sentiment classification task by using word relation, and Hewitt & Manning (2019); Reif et al. (2019) found that the distance between contextual representations contains syntactic information of sentences. Our research focuses on knowledge distillation using context information between words and between layers, and to our best knowledge, we are the first to apply this context information to knowledge distillation.

## 3 SETUP AND BACKGROUND

Most of the recent state-of-the-art language models are stacking Transformer layers which consist of repeated Multi-Head Attentions and Position-wise Feed-Forward Networks.

**Transformer based networks** Given an input sentence with $n$ tokens, $\boldsymbol{X} = [x_1, x_2, \ldots, x_n] \in \mathbb{R}^{d_i \times n}$, most networks (Devlin et al., 2018; Lan et al., 2019; Liu et al., 2019a) utilize the embedding layer to map an input sequence of symbol representations $\boldsymbol{X}$ to a sequence of continuous representations $\boldsymbol{E} = [e_1, \ldots, e_n] \in \mathbb{R}^{d_e \times n}$. Then, each $l$-th Transformer layer of the identical structure takes the previous representations $\boldsymbol{R_{l-1}}$ and produces the updated representations $\boldsymbol{R_l} = [r_{l,1}, r_{l,2}, \ldots, r_{l,n}] \in \mathbb{R}^{d_r \times n}$ through two sub-layers: Multi-head Attention (MHA) and point-wise Feed Forward Network (FFN). The input at the first layer ($l = 1$) is simply $\boldsymbol{E}$. In MHA operation where $h$ separate attention heads are operating independently, each input token $r_{l-1,i}$ for each head is projected into a query $q_i \in \mathbb{R}^{d_q}$, key $k_i \in \mathbb{R}^{d_q}$, and value $v_i \in \mathbb{R}^{d_v}$, typically $d_k = d_q = d_v = d_r/h$. Here, the key vectors and value vectors are packed into the matrix forms $\boldsymbol{K} = [k_1, \cdots, k_n]$ and $\boldsymbol{V} = [v_1, \cdots, v_n]$, respectively, and the attention value $a_i$ and output of each head $o_{h,i}$ are calculated as followed:

$$a_i = \text{Softmax}\left(\frac{\boldsymbol{K}^T \cdot q_i}{\sqrt{d_q}}\right) \quad \text{and} \quad o_{h,i} = \boldsymbol{V} \cdot a_i. \tag{1}$$

The outputs of all heads are then concatenated and fed through the FFN, producing the single word representation $r_{l,i}$. For clarity, we pack attention values of all words into a matrix form $\boldsymbol{A_{l,h}} = [a_1, a_2, .., a_n] \in \mathbb{R}^{n \times n}$ for attention head $h$.

**Knowledge distillation for Transformer** In the general framework of knowledge distillation, teacher network ($T$) with large capacity is trained in advance, and then student network ($S$) with predefined architecture but relatively smaller than teacher network is trained with the help of teacher's knowledge. Specifically, given the teacher parameterized by $\theta_t$, training the student parameterized by $\theta_s$ aims to minimize two objectives: i) the cross-entropy loss $\mathcal{L}^{CE}$ between the output of the student network $S$ and the true label $y$ and ii) the difference of some statistics $\mathcal{L}^D$ between predictions by teacher and student models. Overall, our goal is to minimize the following objective function:

$$\mathcal{L}(\theta_s) = \mathbb{E}_{(\boldsymbol{X}, y)}\left[\mathcal{L}^{CE}\Big(S(\boldsymbol{X}; \theta_s), y\Big) + \lambda \mathcal{L}^D\Big(K^t(\boldsymbol{X}; \theta_t), K^s(\boldsymbol{X}; \theta_s)\Big)\right] \tag{2}$$

where $\lambda$ controls the relative importance between two objectives. Here, $K$ characterizes the knowledge being transferred and can vary depending on the distillation methods, and $\mathcal{L}^D$ is a matching loss function such as $l_1$, $l_2$ or Huber loss. Recent studies on knowledge distillation for Transformer-based BERT can also be understood in this general framework. In particular, DistilBERT (Sanh et al., 2019) matches class predictive probabilities of $S$ and $T$ via Kullback–Leibler divergence: $\mathcal{L}^D_{logit} = \mathbb{KL}(\frac{p_t}{T}, \frac{p_s}{T})$. Patient KD (Sun et al., 2019) matches the individual word representations between $S$ and $T$ in diverse intermediate layers. Recently, in addition to matching word representations, TinyBERT (Jiao et al., 2019) and MobileBERT (Sun et al., 2020) utilize the attention matrix $\boldsymbol{A}$ to transfer the knowledge from $T$ to $S$. More details of these methods are summarized in Appendix A.

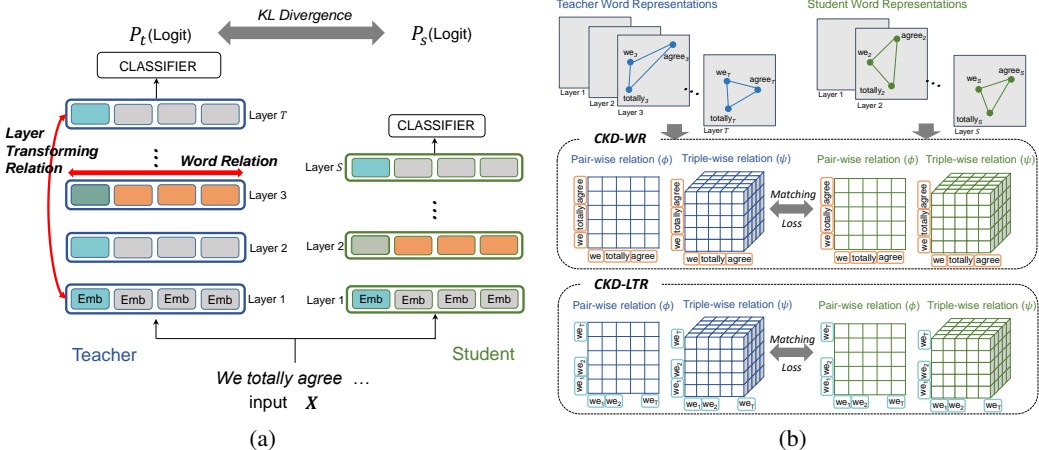

(a)               (b)

Figure 1: **Overview of our contextual knowledge distillation.** (a) In the teacher-student framework, we define the two contextual knowledge, word relation and layer transforming relation which are the statistics of relation across the words from the same layer (orange) and across the layers for the same word (turquoise), respectively. (b) Given the pair-wise and triple-wise relationships of WR and LTR from teacher and student, we define the objective as matching loss between them.

## 4 CONTEXTUAL KNOWLEDGE DISTILLATION

We now present our distillation method that transfers the *structural* or *contextual* knowledge of word representations from teacher to student. Unlike previous methods distilling each word separately, our method transfers structural and contextual information contained in relationships between words or between layers, and provides a more flexible way of constructing embedding space than directly matching representations. The overall structure of our method is illustrated in Figure 1(a). Specifically, we design two key concepts of contextual knowledge from language models: *Word Relation*-based and *Layer Transforming Relation*-based contextual knowledge, as shown in Figure 1(b).

### 4.1 WORD RELATION (WR)-BASED CONTEXTUAL KNOWLEDGE DISTILLATION

Inspired by several previous studies suggesting that neural networks can successfully capture contextual relationships across words (Reif et al., 2019; Mikolov et al., 2013; Pennington et al., 2014), WR-based CKD aims to distill the contextual knowledge contained in the relationships across words at certain layer. The "relationship" across a set of words can be defined in a variety of different ways. Considering the computation burden, our work only focuses on defining it as the sum of *pair-wise* and *triple-wise* relationships. Specifically, for each input $X$ with $n$ words, let $R_l = [r_{l,1}, \cdots r_{l,n}]$ be the word representations at layer $l$ from the language model (it could be teacher or student), as described in Section 3. Then, the objective of WR-based contextual KD is to minimize the following loss:

$$
\begin{aligned}
\mathcal{L}_{\mathrm{CKD-WR}} = &\sum_{(i,j)\in\{1,\dots,n\}^2} w_{ij}\, \mathcal{L}^D\Big(\phi(r_i^s, r_j^s), \phi(r_i^t, r_j^t)\Big) \\
&+\lambda_{\mathrm{WR}} \sum_{(i,j,k)\in\{1,\dots,n\}^3} w_{ijk}\, \mathcal{L}^D\Big(\psi(r_i^s, r_j^s, r_k^s), \psi(r_i^t, r_j^t, r_k^t)\Big)
\end{aligned}
\tag{3}
$$

where $\phi$ and $\psi$ are the functions that define the pair-wise and triple-wise relationships, respectively and $\lambda_{\mathrm{WR}}$ adjust the scales of two losses. Here, we suppress the layer index $l$ for clarity, but the distillation loss for the entire network is simply summed for all layers. Since not all terms in Eq. (3) are equally important in defining contextual knowledge, we introduce the hyperparameters $w_{ij}$ and $w_{ijk}$ to control the weight of how important each pair-wise and triple-wise term is. Determining the values of these hyperparameters is open as an implementation issue, but it can be determined by the locality of words (i.e. $w_{ij} = 1$ if $|i - j| \le \delta$ and 0, otherwise), or by attention information $A$ to focus only on relationship between related words.

While functions $\phi$ and $\psi$ defining pair-wise and triple-wise relationship also have various possibilities, the simplest choices are to use the distance between two words for pair-wise $\phi$ and the angle by three words for triple-wise $\psi$, respectively.

**Pair-wise $\phi$ via distance**   Given a pair of word representations $(r_i, r_j)$ from the same layer, $\phi(r_i, r_j)$ is defined as follows: $\phi(r_i, r_j) = \|r_i - r_j\|_2$.

**Triple-wise $\psi$ via angle**   Triple-wise relation captures higher-order structure and provides more flexibility in constructing contextual knowledge. However, as an expense of using higher-order information, additional constraints such as computation or memory costs should be considered. One of the simplest forms for $\psi$ is the angle, which is calculated as

$$\psi(r_i, r_j, r_k) = \cos \angle(r_i, r_j, r_i) = \left\langle \frac{r_i - r_j}{\|r_i - r_j\|_2}, \frac{r_k - r_j}{\|r_k - r_j\|_2} \right\rangle \tag{4}$$

where $\langle \cdot, \cdot \rangle$ denotes the dot product between two vectors.

Despite its simple form, efficiently computing the angles in Eq. (4) for all possible triples out of $n$ words requires storing all relative representations $(r_i - r_j)$ in a $(n, n, d_r)$ tensor[1]. This incurs an additional memory cost of $\mathcal{O}(n^2 d_r)$. In this case, using locality for $w_{ijk}$ in Eq. (3) mentioned above can be helpful; by considering only the triples within a distance of $\delta$ from $r_j$, the additional memory space required for efficient computation is $\mathcal{O}(\delta n d_r)$, which is beneficial for $\delta \ll n$. In the experimental section, we show that measuring angles in local window does not hurt the performance to some extent.

## 4.2   LAYER TRANSFORMING RELATION (LTR)-BASED CONTEXTUAL KNOWLEDGE DISTILLATION

The second structural knowledge that we propose to capture is on *"how each word is transformed as it passes through the layers"*. Transformer-based language models are composed of a stack of identical layers and thus generate a set of representations for each word, one for each layer, with more abstract concept in the higher hierarchy. Hence, LTR-based contextual KD aims to distill the knowledge of how *each* word develops into more abstract concept within the hierarchy. Toward this, given a set of representations for a single word in $L$ layers, $[r_{1,w}^s, \cdots, r_{L,w}^s]$ for student and $[r_{1,w}^t, \cdots, r_{L,w}^t]$ for teacher (Here we abuse the notation and $\{1, \ldots, L\}$ is not necessarily the entire layers or student of teacher. It is the index set of layers for which we want to distill the knowledge; this time, we will suppress the word index below), the objective of LTR-based contextual KD is to minimize the following loss:

$$\mathcal{L}_{\text{CKD-LTR}} = \sum_{(l,m)\in\{1,\ldots,L\}^2} w_{lm} \, \mathcal{L}^D\Big(\phi(r_l^s, r_m^s), \phi(r_l^t, r_m^t)\Big)$$

$$+ \lambda_{\text{LTR}} \sum_{(l,m,o)\in\{1,\ldots,L\}^3} w_{lmo} \, \mathcal{L}^D\Big(\psi(r_l^s, r_m^s, r_o^s), \psi(r_l^t, r_m^t, r_o^t)\Big). \tag{5}$$

where $\lambda_{\text{LTR}}$ again adjust the scales of two losses. Here, the composition of Eq. (5) is the same as Eq. (3), but only the objects for which the relationships are captured have been changed from word representations in one layer to representations for a single word in layers. That is, the relationships among representations for a word in different layers can be defined from distance or angle as in Eq. (4): $\phi(r_l, r_m) = \|r_l - r_m\|_2$ and $\psi(r_l, r_m, r_o) = \langle \frac{r_l - r_m}{\|r_l - r_m\|_2}, \frac{r_o - r_m}{\|r_o - r_m\|_2} \rangle$.

**Alignment strategy**   When the numbers of layers of teacher and student are different, it is important to determine which layer of the student learns information from which layer of the teacher. Previous works (Sun et al., 2019; Jiao et al., 2019) resolved this *alignment* issue via the *uniform (i.e. skip) strategy* and demonstrated its effectiveness experimentally. For $L_t$-layered teacher and $L_s$-layered student, the layer matching function $f$ is defined as

$$f(\text{step}_s \times t) = \text{step}_t \times t, \quad \text{for } t = 0, \ldots, g$$

where $g$ is the greatest common divisor of $L_t$ and $L_s$, $\text{step}_t = L_t/g$ and $\text{step}_s = L_s/g$.

---

[1]From the equation $\|r_i - r_j\|_2^2 = \|r_i\|_2^2 + \|r_j\|_2^2 - 2\langle r_i, r_j \rangle$, computing the pair-wise distance with the right hand side of equation requires no additional memory cost.

Table 1: Comparisons against state-of-the-art distillation methods. For a fair comparison, all students are 6/768 BERT models, distilled by BERT$_{\text{BASE}}$ (12/768) teachers. Other results are as reported by their authors. Results of development set are averaged over 5 runs and the best one of them are used for test server evaluation. "-" means the result is not reported.

|  | Model | CoLA (Mcc) | MNLI-(m/-mm) (Acc) | SST-2 (Acc) | QNLI (Acc) | MRPC (F1) | QQP (Acc) | RTE (Acc) | STS-B (Spear) | Score |
|---|---|---|---|---|---|---|---|---|---|---|
| dev | BERT$_{\text{BASE}}$ (Teacher) | 60.4 | 84.8/84.6 | 94.0 | 91.8 | 90.3 | 91.4 | 70.4 | 89.5 | 84.1 |
|  | DistilBERT | 42.5 | 81.6/81.1 | 92.7 | 85.5 | 88.3 | 90.6 | 60.0 | 85.0 | 78.6 |
|  | PD | - | 82.5/83.4 | 91.1 | 89.4 | 89.4 | 90.7 | 66.7 | - | - |
|  | TinyBERT (w/ DA) | 54.0 | 84.5/84.5 | 93.0 | 91.1 | 90.6 | 91.1 | **73.4** | **89.6** | 83.5 |
|  | **CKD** | 52.8 | 83.9/84.4 | 93.3 | 90.5 | 89.6 | 90.9 | 67.3 | 89.0 | 82.4 |
|  | **CKD** (w/ DA) | **57.9** | **84.8/85.0** | **93.8** | **91.7** | **90.8** | **91.6** | 70.1 | **89.6** | **83.9** |
| test | BERT$_{\text{BASE}}$ (Teacher) | 53.8 | 84.7/83.8 | 93.8 | 90.9 | 88.4 | 89.2 | 67.6 | 85.2 | 81.9 |
|  | PKD | - | 81.5/81.0 | 92.0 | 89.0 | 85.0 | 88.9 | 65.5 | - | - |
|  | PD | - | 82.8/82.2 | 91.8 | 88.9 | 86.8 | 88.9 | 65.3 | - | - |
|  | TinyBERT (w/ DA) | 51.1 | **84.6**/83.2 | 93.1 | 90.4 | 87.3 | 89.1 | **70.0** | 83.7 | 81.4 |
|  | **CKD** | 50.5 | 84.1/82.8 | 93.1 | 90.0 | 87.3 | 89.1 | 65.1 | 82.4 | 80.5 |
|  | **CKD** (w/ DA) | **52.8** | **84.6/84.0** | **93.5** | **90.7** | **88.0** | **89.7** | 66.2 | **83.8** | **81.5** |

**Final objective** The distillation objective aims to supervise the student network with the teacher's knowledge. Multiple distillation loss functions can be used during training, either alone or together. The proposed CKD can also be combined simply with the vanilla distillation loss such as class probability matching (Hinton et al., 2015) as an additional term. In that case, our distillation objective is as follows:

$$\mathcal{L} = (1 - \alpha)\mathcal{L}_{\text{CE}} + \alpha\mathcal{L}_{logit}^{\text{D}} + \lambda_{\text{CKD}}\Big(\mathcal{L}_{\text{CKD-LTR}} + \mathcal{L}_{\text{CKD-WR}}\Big) \quad (6)$$

where $\alpha$ and $\lambda_{\text{CKD}}$ is a tunable parameter to balance the loss terms.

## 5 EXPERIMENTS

We validate our CKD using the Stanford Question Answer Dataset (SQuAD) and General Language Understanding Evaluation (GLUE) benchmark (Wang et al., 2018) which consists of 9 natural language understanding tasks such as Natural Language Inference, Sentiment Classification, and Paraphrase Similarity Matching. Following the Devlin et al. (2018), all datasets except the WNLI dataset are used for experiments. We first compare our method with the state-of-the-art distillation objectives in training 6-layer BERT student network. We then report on the performance gains achieved by our method for BERT architectures of various sizes, including the recent MobileBERT (Sun et al., 2020). Finally, we analyze the effect of each component of our CKD and the impact of leveraging locality $\delta$ for $w_{ijk}$ in Eq. (3).

**Setup** In all our experiments, to avoid time-consuming BERT pre-training process, we apply our CKD on top of pre-trained BERT models of various sizes, released by Turc et al. (2019). Also, in order to reduce the hyperparameter search cost of our method, we do not allow full degrees of freedom for $\lambda_{\text{WR}}$ and $\lambda_{\text{LTR}}$, and search only within the range where they have the same value. Hence, we only have two hyperparameters to tune ($\lambda_{\text{WR}}$ and $\lambda_{\text{CKD}}$) in CKD. For the importance weight of each pair-wise and triple-wise terms, we leverage the locality of words, in that $w_{ij} = 1$ if $|i - j| \leq \delta$ and 0, otherwise. For this, we select the $\delta$ in (10-15). More details are provided in Appendix D

### 5.1 COMPARISONS AGAINST STATE-OF-THE-ART DISTILLATION METHODS

To verify the effectiveness of our CKD objective, we compare the performance with the current state-of-the-art distillation methods for BERT compression. We have four baselines objectives: DistilBERT (Sanh et al., 2019), PKD (Sun et al., 2019), PD (Turc et al., 2019) and TinyBERT (Jiao et al., 2019). Following the standard setup in baselines, we use the BERT$_{\text{BASE}}$ (12/768)[2] as the teacher and 6-layer BERT (6/768) as the student network. Therefore, the student models used in all baselines and ours have the same number of parameters (67.5M) and FLOPs (10878M).

---

[2]In notation $(a/b)$, $a$ means the number of layers and $b$ denotes a hidden size in intermediate layers.

Table 2: Comparisons against state-of-the-art distillation methods on the SQuAD 1.1v dataset (EM/F1 on dev set). For a fair comparison, all students are 6/768 BERT models, distilled by BERT$_{BASE}$ (12/768) teachers. The results of PKD and TinyBERT are as reported by Jiao et al. (2019) and the result of DistilBERT is as reported by the author (Sanh et al., 2019).

| Model | #Params | #FLOPs (Speed up) | SQuAD 1.1v EM | F1 |
|---|---|---|---|---|
| BERT$_{BASE}$ (Teacher) | 110M | 21754M (1.00x) | 81.3 | 88.6 |
| PKD | 67.5M | 10878M (2.00x) | 77.1 | 85.3 |
| DisitlBERT | 67.5M | 10878M (2.00x) | 79.1 | 86.9 |
| TinyBERT | 67.5M | 10878M (2.00x) | 79.7 | 87.5 |
| **CKD** | 67.5M | 10878M (2.00x) | **81.8** | **88.7** |

Table 3: Experimental results with the recently proposed MobileBERT. Following the author, a baseline is trained without distillation for downstream tasks. Results are averaged over 5 runs on the development set.

| Model | CoLA (Mcc) | MNLI-(m/-mm) (Acc) | SST-2 (Acc) | QNLI (Acc) | MRPC (F1) | QQP (Acc) | RTE (Acc) | STS-B (Spear) | Score |
|---|---|---|---|---|---|---|---|---|---|
| MobileBERT (Sun et al., 2020) | 54.0 | 83.4/83.8 | 92.1 | 91.2 | **90.8** | 90.5 | 64.7 | 88.1 | 82.1 |
| MobileBERT (w/ **CKD**) | **54.8** | **84.1/84.3** | **92.3** | **91.4** | 90.6 | **90.6** | **67.1** | **88.2** | **82.6** |

Table 1 summarizes results both for the development set and test set of GLUE datasets and illustrates how much better our CKD performs than the recent distillation methods. We present two cases depending on whether data augmentation (following Jiao et al. (2019)) is used for training or not. Without the data augmentation, CKD surpasses the DistilBERT, PD, and PKD by a large margin for all datasets in GLUE. For the case with data augmentation, CKD outdoes TinyBERT in all scores except for the RTE development and test set, and in terms of the average score of all development datasets, CKD has a 0.4 %p improvement over TinyBERT that transfers both individual word representations and attention matrices. Moreover, for some datasets including MNLI and QQP, student network trained with CKD even surpass the performance of teacher with the help of data augmentation. This supports the claim that CKD is a more flexible way of constructing embedding space than directly matching representations. In Table 2, CKD exhibits the best performance on the SQuAD 1.1v dataset with the same student architecture. Our CKD has a 2.1%p improvement over TinyBERT, which is the current state-of-the-art knowledge distillation method.

## 5.2 BOOSTING THE STATE-OF-THE-ART THIN BERT ARCHITECTURE

Recently, MobileBERT (Sun et al., 2020) is proposed as a new type of thin architecture of BERT, trained with task-agnostic distillation. MobileBERT also employs the distillation policy and transfers individual word representations and attention matrices to further improve its performance. However, it applies the distillation loss only to the pre-training process and modifies the original architectures of teacher and student BERT models, thus making direct comparisons difficult. Instead, given the same pre-trained MobileBERT[3], we evaluate the performance of fine-tuning on GLUE datasets using our CKD compared to the method originally in Sun et al. (2020). For this experiment, following Sun et al. (2020), we search the additional hyperparameters in a search space including different batch sizes (16/32/48), learning rates ($(1 - 10) \times e^{-5}$) and the number of epochs (2-10). The results are summarized in Table 3. Fine-tuning with CKD distillation boosts the MobileBERT's original performance on all datasets except MRPC, which is a relatively small dataset.

## 5.3 EFFECT OF MODEL SIZE FOR CKD

For the knowledge distillation with the purpose of network compression, it is essential to work well in more resource-scarce environments. To this end, we further evaluate our method in two settings.

First, we select small BERTs of a certain size and compare naive fine-tuning and our CKD with them on the 8 GLUE dataset. In this experiment, we use the BERT$_{BASE}$ as the teacher and consider the following student models: BERT$_{MINI}$ (6/256), BERT$_{SHALLOW}$ (12/256 )and BERT$_{SMALL}$ (4/512).

---

[3]We use the pre-trained MobileBERT released in the official repository

Table 4: Experiment results on the effect of model sizes for CKD and TinyBERT results are as reported in their papers. The results are evaluated on the test set of GLUE official benchmark. Detailed component of model sizes are provided in the appendix C. † marks our runs with the official code.

| Model | Objective | #Params | #FLOPs (Speed up) | CoLA (Mcc) | MNLI (-m/-mm) (Acc) | SST-2 (Acc) | QNLI (Acc) | MRPC (F1) | QQP (Acc) | RTE (Acc) | STS-B (Spear) | Score |
|---|---|---|---|---|---|---|---|---|---|---|---|---|
| BERT$_{BASE}$ (Reported) | - | 110.1M | 21754M (1.00x) | 52.1 | 84.6/83.4 | 93.5 | 90.5 | 88.9 | - | 66.4 | 85.8 | - |
| BERT$_{BASE}$ (Teacher) | - | | | 53.8 | 84.7/83.8 | 93.8 | 90.9 | 88.4 | 89.2 | 67.6 | 85.2 | 81.9 |
| BERT$_{SMALL}$ | Naive | 29.1M | 3324M (6.54x) | 38.0 | 75.1/74.8 | 89.3 | 84.5 | 85.2 | 84.7 | 64.2 | 77.3 | 74.8 |
| | **CKD** | | | 48.3 | 82.4/81.8 | 92.9 | 88.3 | 87.0 | 89.1 | 64.6 | 82.6 | **79.7** |
| BERT$_{SHALLOW}$ | Naive | 17.6M | 2419M (8.99x) | 39.3 | 76.5/76.7 | 88.5 | 84.2 | 85.4 | 85.4 | 63.5 | 77.5 | 75.2 |
| | **CKD** | | | 51.9 | 82.6/81.9 | 93.1 | 88.1 | 87.5 | 89.0 | 65.5 | 83.0 | **80.3** |
| TinyBERT (Reported) | - | 14.5M | 1167M (18.64x) | 43.3 | 82.5/81.8 | 92.6 | 87.7 | 86.4 | 87.7† | 62.9 | 79.9 | 78.3 |
| BERT$_{MINI}$ | Naive | 12.5M | 1210M (17.98x) | 29.7 | 74.7/74.2 | 86.7 | 83.0 | 82.7 | 83.6 | 62.7 | 73.3 | 72.3 |
| | **CKD** | | | 45.7 | 81.7/80.8 | 92.9 | 87.3 | 86.9 | 88.3 | 63.9 | 81.7 | **78.8** |

We also present the number of parameters and floating-point operations (FLOPs) to measure the computational complexity of student models regardless of the operating environments. Speed up in Table 4 is also calculated based on the FLOPs. To observe just how much distillation alone improves performance, we use data augmentation both for the baseline and our method.

The results are summarized in Table 4. The results illustrate that our CKD consistently exhibits significant improvements in the performance compared to naive fine-tuning on all datasets. For all downsized student models, the average score also improved by more than 4.9% point. Moreover, comparing BERT$_{MINI}$ and TinyBERT, we observe that the BERT$_{MINI}$ trained with the CKD achieves the higher average score while it has few model parameters with comparable but slightly higher FLOPs.

Second, we perform extensive analyses of the effect of model sizes. We experiment on the SST-2 dataset with various model sizes (2∼10 layers and 128∼768 intermediate dimensions) and compare the CKD and the two baselines: Naive fine-tuning and class probability matching (Logit KD). As above, we use the BERT$_{BASE}$ as the teacher and the BERT$_{SMALL}$ as the student. The results are averaged over 5 runs on the development set. As illustrated in Figure 2, our CKD consistently performs better than the two baselines in most model sizes. In addition, we observe that as the layer deepens or the embedding size increases, the gap in the performance between CKD and baselines increases.

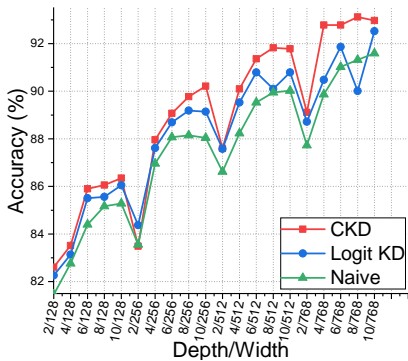

Figure 2: In L/H, L denotes the number of hidden layers and H does the embedding size. Listing of x-axis is width-first.

## 5.4 ABLATION STUDIES

We provide additional ablation studies to analyze the impact of each component of the CKD: Word relation, Layer transforming relation, Pair-wise and Triple-wise relationship and the introduced locality ($w_{i,j} = \delta$) in Eq. (3) as the weight of how important each pair-wise and triple-wise term is. For these studies, we fix the student network with BERT$_{SMALL}$ and report the results as an average over 5 runs on the development set.

**Impact of each component of CKD** The proposed CKD transfers the word relation based and layer transforming relation based contextual knowledge. To isolate the impact of them, we experiment successively removing each piece of our objective. Table 5 summarizes the results, and we observe that WR and LTR brings a considerable performance gain, verifying their individual effectiveness. As shown in Table 5, both pair-wise and angle-wise relationship help improve performance, respectively, and the best performance is achieved when all components are applied together.

**Locality as the importance of relation terms** We introduced the additional weights ($w_{ij}$, $w_{ijk}$) in Eq. (3) for CKD-WR (and similar ones for CKD-LTR) to control the importance of each pair-wise and triple-wise term and suggested to use the locality for them as one possible way. Here, we verify the effect of locality by increasing the local window size ($\delta$) on the SST-2 and QNLI datasets. In particular, the triple-wise relation in Eq. (3) requires a large amount of additional memory as discussed

Table 5: Ablation study using the subset of GLUE about the impact of each component of CKD.

| CKD − WR | | CKD − LTR | | SST-2 | MRPC | QNLI |
|---|---|---|---|---|---|---|
| Pair-wise | Triple-wise | Pair-wise | Triple-wise | (Acc) | (F1) | (Acc) |
| - | - | - | - | 89.3 | 86.2 | 85.9 |
| ✓ | ✓ | - | - | 90.1 | 86.9 | 87.2 |
| - | - | ✓ | ✓ | 89.6 | 88.4 | 86.8 |
| ✓ | - | ✓ | - | **90.6** | 88.1 | 87.1 |
| - | ✓ | - | ✓ | 90.2 | 88.3 | 86.6 |
| ✓ | ✓ | ✓ | ✓ | **90.6** | **89.2** | **87.4** |

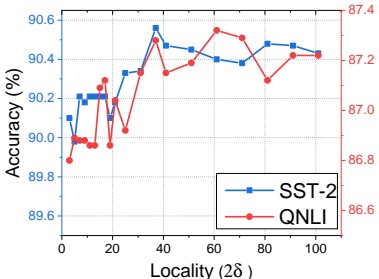

Figure 3: Ablation study about the effect of locality.

in Section 4.1, so using this locality for their weights is a practical solution. Hence, we investigate how reducing memory usage using this locality for them affects overall performance. The result is illustrated in Figure 3. We observe that as the local window size increases, the performance improves, but after some point, the performance is almost saturated. This result exhibits that computing angles in local window does not hurt the performance to some extent.

## 6 CONCLUSION

We proposed a novel distillation strategy, contextual knowledge distillation (CKD), that leverages contextual information efficiently based on word relation and layer transforming relation. On the standard GLUE benchmark, it performs better than existing state-of-the-art methods and even surpasses the teacher's performance on several datasets with the help of data augmentation. We also showed that our method boosts the performance of MobileBERT, and that it performs consistently well for student models of various sizes. Through the ablation studies, it is confirmed that all components of the proposed CKD method are helpful in performance. As future work, we plan to define the weight between each relation (i.e., $w$) continuously using attention matrices, not identically or discretely using locality.

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

Table 6: Overview of distillation objectives used for language model compression and their constraint on architecture. Although TinyBERT acquires flexibility on the embedding size, using an additional parameter, for attention matrices matching, the number of attention heads of the teacher and student must be the same. Although Sun et al. (2020) experiment with various sizes of models, they train the modified teacher architecture to distill the word representation and attention matrix.

| | Knowledge Distillation Objectives | Constraint |
|---|:---:|:---:|
| DistilBERT (Sanh et al., 2019) | $\sum_{i=1}^{n} \cos(r_{l,i}^t, r_{l,i}^s), \mathcal{L}_{Logit}^{\mathrm{D}}$ | Embedding size |
| PKD (Sun et al., 2019) | $\sum_{i=1}^{n} \Big[ \mathrm{MSE}\big(\frac{r_{l,i}^t}{\|r_{l,i}^t\|_2} - \frac{r_{l,i}^s}{\|r_{l,i}^s\|_2}\big)\Big], \mathcal{L}_{Logit}^{\mathrm{D}}$ | Embedding size |
| PD (Turc et al., 2019) | $\mathcal{L}_{Logit}^{\mathrm{D}}$ | - |
| TinyBERT (Jiao et al., 2019) | $\sum_{i=1}^{n} \Big[ \mathrm{MSE}(r_{l,i}^t - W_r r_{l,i}^s)\Big], \sum_{h=1}^{H} \Big[ \mathrm{MSE}(\boldsymbol{A}_{l,h}^t - \boldsymbol{A}_{l,h}^s)\Big], \mathcal{L}_{Logit}^{\mathrm{D}}$ | Attention head |
| Mobile-BERT (Sun et al., 2020) | $\sum_{i=1}^{n} \Big[ \mathrm{MSE}(r_{l,i}^t - r_{l,i}^s)\Big], \sum_{h=1}^{H} \Big[ \mathbb{KL}\big(\boldsymbol{A}_{l,h}^t, \boldsymbol{A}_{l,h}^s\big)\Big], \mathcal{L}_{Logit}^{\mathrm{D}}$ | Embedding size Attention head |

# A  EXPLANATION OF PREVIOUS METHOD: LEVERAGING INDIVIDUAL WORD REPRESENTATION

Table 6 summarizes the knowledge distillation objectives of previous methods and their constraints. **DistilBERT** (Sanh et al., 2019) uses logit distillation loss (Logit KD), masked language modeling loss, and cosine loss between the teacher and student word representations in the learning process. The cosine loss serves to align the directions of the hidden state vectors of the teacher and student. Since the cosine of the two hidden state vectors is calculated in this process, they have the constraint that the embedding size of the teacher and the student model must be the same.

**PKD** (Sun et al., 2019) transfers teacher knowledge to the student with Logit KD and patient loss. The patient loss is the mean-square loss between the normalized hidden states of the teacher and student. To calculate the mean square error between the hidden states, they have a constraint that the dimensions of hidden states must be the same between teacher and student.

**PD** (Turc et al., 2019) raised the importance of the pre-training process. By pre-training without distillation and employing only logit kd in the fine-tuning process, they achieved results comparable to other methods. Since only the logit KD is employed, there is no restriction on architecture.

**TinyBERT** (Jiao et al., 2019) uses additional loss that matches word representations and attention matrices between the teacher and student. Although they acquire flexibility on the embedding size, using an additional parameter, since the attention matrices of the teacher and student are matched through mean square error loss, the number of attention heads of the teacher and student must be the same.

**MobileBERT** (Sun et al., 2020) utilizes a similar objective with TinyBERT (Jiao et al., 2019) for task-agnostic distillation. However, since they match the hidden states with $l2$ distance, and attention matrices with $\mathcal{KL}$ divergence between teacher and student, they have restrictions on the size of hidden states and the number of attention heads.

The methods introduced in Table 6 have constraints by their respective knowledge distillation objectives. However, our CKD method which utilizes the relation statistics between the word representations (hidden states) has the advantage of not having any constraints on student architecture.

## B EXPLANATION OF DATA AUGMENTATION

---

**Algorithm 1** Data Augmentation

---

**Input:** sentence $x$, hyperparameters $N$, $M$. $p_t$
**Output:** augmented data $Y$ (list of sentence)

---

**for** $i = 1$ **to** $len(x)$ **do**
   **if** $x[i]$ is a single-piece word **then**
      $x_{masked} \leftarrow x$
      Replace $x_{masked}[i]$ with '[MASK]' token
      $candidates[i] \leftarrow$ M most-likely words predicted for '[MASK]' by $BERT(x_{masked})$
   **else**
      $candidates[i] \leftarrow$ M similar words of $x[i]$ from GloVe
   **end if**
**end for**
Append $x$ to $Y$
$n \leftarrow 0$
**while** $n < N$ **do**
   $x_{candidate} \leftarrow x$
   **for** $i = 1$ **to** $len(x)$ **do**
      Sample $p \sim U(0, 1)$
      **if** $p \leq p_t$ **then**
         Replace $x_{candidate}[i]$ with a randomly picked word from $candidate[i]$
      **end if**
   **end for**
   Append $x_{candidate}$ to $Y$
   $n \leftarrow n + 1$
**end while**

---

In this paper, for a fair comparison with TinyBERT (Jiao et al., 2019), data augmentation is used. We describe the data augmentation method performed on the GLUE benchmark datasets in this section. The overall process is summarized in algorithm 1. Data augmentation consists of the processes selecting replaceable candidates for each word in a sentence and generating new sentences using the candidates.

First, replaceable candidate words (the number of candidate words is $M$) are selected for each word in the sentence. In this process, the method of selecting candidates is divided according to whether the word is a single-piece word or not. If the word is a single-piece word, $M$ candidate words are selected in a way that predicts the '[MASK]' token replaced in same position through the language model. We use the BERT$_{\text{Large}}$ for the language model. If the word is not a single-piece word, the $M$ candidate words are selected using GloVe pre-trained embedding through cosine similarity with the target word. After selecting candidate words in the sentence, a total of $N$ augmented sentences are generated. For each word in the sentence, it is randomly selected and changed among $M$ candidate words with a probability of $p_t$. If this process is repeated $N$ times, a total of $N + 1$ augmented sentences including the original sentence are created.

In the case of datasets consisting of a pair of sentences such as QQP or MNLI in GLUE benchmark datasets, each sentence is augmented with $N$ sentences. In other words, since the paired sentence is attached to $N + 1$ augmented sentences as originally, a total of $2(N + 1)$ sentences are generated. In this study, we use $M = 15$, $p_t = 0.4$, and $N = 20$ for all datasets except the QQP and MNLI which are the original large datasets. For the QQP and MNLI, we use $N = 2$.

## C EXPLANATION OF VARIOUS MODEL SIZE

Table 7: Details of architecture used for our experiments

| Model | Layers | Hidden Size | Feed-forward Size | Attention Heads | Model Size |
|---|---|---|---|---|---|
| BERT$_{BASE}$ | 12 | 768 | 3072 | 12 | 110.1M |
| BERT$_{SMALL}$ | 4 | 512 | 2048 | 8 | 29.1M |
| MobileBERT (Sun et al., 2020) | 24 | 512 | 512 | 4 | 25.3M |
| BERT$_{SHALLOW}$ | 12 | 256 | 1024 | 4 | 17.6M |
| TinyBERT (Jiao et al., 2019) | 4 | 312 | 1200 | 12 | 14.5M |
| BERT$_{MINI}$ | 6 | 256 | 1024 | 4 | 12.5M |

In this section, we describe the various models used in our experiments. Since pre-training each model of various sizes costs a lot, pre-trained models of various sizes provided by PD (Turc et al., 2019) are used for our experiments. In addition, MobileBERT introduce the Inverted-Bottleneck BERT. Excluding BERT$_{BASE}$, a total of 3 student models are used, and each model name can be different from Turc et al. (2019). Table 7 summarizes the number of layers, hidden size, feed-forward size, attention heads, and model size of each model.

## D EXPERIMENT SETTING

This section introduces the experimental setting in details. Our contextual knowledge distillation proceeds in the following order. First, from pre-trained large BERT, task-specific fine-tuning is conducted to serve as teacher. Then, prepare the pre-trained small-size architecture which serve as student. In this case, pre-trained models of various model sizes provided by Turc et al. (2019) are employed. Finally, task-specific distillation with our CKD is performed.

We implemented with PyTorch framework and huggingface's transformers package (Wolf et al., 2019). To reduce the hyperparameters search cost, $\lambda_{WR}$ in Eq. (3) and $\lambda_{LTR}$ in Eq. (5) are used with same value. For the important weights of pair-wise and triple-wise terms, the locality is applied only to the importance weight $w$ of the word relation (WR)-based CKD loss. The importance weight $w$ of the layer transforming relation (LTR)-based CKD loss is set to 1. In this paper, we report the best result among the following values to find the optimal hyperparameters of each dataset:

- Batch size : 32
- Learning rate : (3, 4, 5) 3e-5, 4e-5, 5e-5
- Number of epochs : 4, 8
- Alpha ($\alpha$) : 0.7, 0.9
- Temperature ($T$) : 1, 2, 3, 4, 5
- $\lambda_{WR}, \lambda_{LTR}$ : 0.1, 1, 10, 100, 1000
- $\lambda_{CKD}$ : 1, 10, 100, 1000

