# OpenReview forum: "Contextual Knowledge Distillation for Transformer Compression"
_ICLR.cc/2021/Conference — Reject_

### Official Review · AnonReviewer3 · 2020-10-26
**A paper that proposes two new distillation objective based on word relations and layer transforming relations that outperform previous distillation methods.**

**Rating:** 6
**Confidence:** 5

**Review:**

This paper proposes two new distillation objective, word relations and layer transforming relations. Word relations constrain the pairwise/triplet relations of embeddings at each layer to be closer to the teacher network. The layer transforming relations constrain that the pairwise/triplet relations of embeddings between different layers should match the teacher network.

pros:

1. The methods in this paper consider the pairwise or higher order (i.e. triplet)  relations to constrain  the student embeddings, while previous methods usually consider embeddings separately. The methods shall provide more constrained information from the teacher network.

2. Comparison with previous methods, ablation study and other experiments like model sizes, etc demonstrate the effectiveness of the proposed method. In some cases, the student network even outperforms the teacher network (more explanation about this might be needed).

Cons (or questions):

1. Why the angle-based method is adopted, instead of other methods (e.g. the maximum/average distance between the triplet)? Is there any experiments studying the effect of the choice of these functions?

2. Previous methods sometimes use a short network (fewer layers) or thin network (smaller hidden sizes). As a result, I am not sure whether the number of parameters are comparable when comparing to the baselines. Could the authors also show the number of parameters of previous methods and their own methods?

Typo:
, etc. (Devlin et al., 2018; Lan et al., 2019; Liu et al., 2019a; Raffel et al., 2019; Yang et al., 2019): citations should be put before the punctuation.

---

> ### Author Response · Authors · 2020-11-18
> **Response to Reviewer #3**
>
> Thank you for the constructive review. We address your concerns below:
>
> **Q1: [In some cases, the student network even outperforms the teacher network (more explanation about this might be needed).]**
>
> - As research in the field of knowledge distillation advances, there are several cases that student networks outperform a teacher with well-designed knowledge. Table 1 below presents the case examples. In addition, for the case of CKD (w/ DA) of Table 1 in the paper, data augmentation was used only in the distillation process. It may be helped to boost student performance.
>
>
>
> Table 1. The case examples show that the student network can perform better with fewer parameters
> than the teacher network. All student models have fewer parameters than teacher models.
>
> | Paper           |      $\hskip0.6cm$  Task (dataset)       | Metric | Teacher (model)       | Student (model)         |
> |-----------------|:--------------------------:|:------:|-----------------------|-------------------------|
> | TinyBERT [1]    |        NLU (RTE/QQP)       | Acc/F1 | 67.0/71.1 (BERT base) | **70.0/71.6** (6-BERT)  |
> | CRD [2]         | Classification (CIFAR-100) |   Acc  | 75.6 (WRN-40-2)       | **76.1** (ShuffleNetV1) |
> | PAD-L2 [3]      | Classification (CIFAR-100) |   Acc  | 63.5 (ResNet18)       | **65.6** (MobileNetV2)  |
> | SuperMix+KD [4] | Classification (CIFAR-100) |   Acc  | 74.6 (VGG13)          | **76.0** (VGG8)         |
> | CD+GKD+EDT [5]  | Classification (CIFAR-100) |   Acc  | 80.9 (ResNet152)      | **81.4** (ResNet50)     |
>
> ---
>
> **Q2: [Why the angle-based method is adopted, instead of other methods (e.g. the maximum/average distance between the triplet)? Is there any experiments studying the effect of the choice of these functions?]**
>
> - Thank you for the helpful suggestion. As the reviewer pointed out, the form of measuring pairwise and triple-wise relationships in our method is open. Following the reviewer’s suggestion, we perform an additional study on the choice of the triplet relation functions. In this experiment, we use the subset of GLUE and BERT(small) architecture. As shown in Table 2 below, the angle-based relation consistently shows the best performance compared to others on 4 datasets. We think that knowledge in the other methods (Maximum/Average distance between the triplet) is implicitly contained in the pair-wise distance which is already modeled via the pair-wise relation.
>
>
>
> Table 2. Ablation study for the choice of triplet relation method. For this experiment, we do not
> use data augmentation. The results are averaged over 5 runs on the development set.
>
> | Method | CoLA | MNLI-m | MNLI-mm | MRPC | QNLI |
> |-|-|-|-|-|-|
> | CKD (Pairwise only) | 34.7(±0.43) | 81.3(±0.11) | 81.4(±0.16) | 88.1(±0.28) | 87.1(±0.14) |
> | CKD (Pairwise & AvgDist) | 35.7(±0.31) | 81.7(±0.07) | 81.6(±0.13) | 88.7(±0.22) | 87.2(±0.09) |
> | CKD (Pairwise & MaxDist) | 35.4(±0.38) | 81.4(±0.09) | 81.1(±0.16) | 88.1(±0.19) | 86.9(±0.10) |
> | CKD (Pairwise & Angle) | **37.9(±0.42)** | **82.4(±0.07**) | **81.9(±0.11)** | **89.2(±0.24)** | **87.4(±0.09)** |
>
> ---
>
> **Q3: [Previous methods sometimes use a short network (fewer layers) or thin network (smaller hidden sizes). ... Could the authors also show the number of parameters of previous methods and their own methods?]**
>
> - To compare the other distillation objectives fairly in **Table 1**of the paper, we follow the **standard setup**in baselines which use the 6-layer BERT as the student network. Therefore, **the student models used in all baselines and our methods have the same number of parameters (67.5M) and FLOPs (10878M).**  In Table 4 of the paper where we shows the effect of our CKD on various sized network architectures, the number of parameters for all architectures are also reported. For more clarity, in the revision, we additionally report FLOPs and Speed up to validate the benefits of inference times for each student architecture.
>
> ---
>
> **Q4: [Typo: , etc. (Devlin et al., 2018; Lan et al., 2019; Liu et al., 2019a; Raffel et al., 2019; Yang et al., 2019): citations should be put before the punctuation.]**
>
> - Thank you for letting us know that. We correct this typo error in the paper.
>
> **References**
> [1] Jiao, et.al. “TinyBERT: Distilling BERT for Natural Language Understanding”, EMNLP 2020
> [2] Tian et al. “Contrastive Representation Distillation”, ICLR 2020
> [3] Zhang et al. “Prime-Aware Adaptive Distillation”, ECCV 2020
> [4] Dabouei et al. “SuperMix: Supervising the Mixing Data Augmentation”, arXiv 2020
> [5] Zhou et al. “Channel Distillation: Channel-Wise Attention for Knowledge Distillation”, arXiv 2020

---

> > ### Comment · AnonReviewer3 · 2020-11-23
> > **Thanks for the response!**
> >
> > Thanks for the response from the authors. The concerns are well addressed.

---

> > > ### Author Response · Authors · 2020-11-24
> > > **Thanks for your reviews.**
> > >
> > > We sincerely thank you for your time and effort in reviews.
> > > We believe that our paper gets much stronger and clearer through this rebuttal.
> > > Thank you again for your constructive suggestions and please keep safe!

---

### Official Review · AnonReviewer1 · 2020-10-27
**Well written paper on KD for Transformer-based models, the experiments can be improved**

**Rating:** 5
**Confidence:** 4

**Review:**

This paper presents an interesting knowledge distillation method based on a newly defined contextual knowledge of transformer-based models. The proposed contextual knowledge models the pair-wise or triple-wise relations across BERT-based contextual representations, based on which the local structures between Teacher and Student models are encouraged to be well aligned.

The main contribution of this work lies in the newly proposed two types of contextual knowledge: Word Relation and Layer Transforming Relation. By using this new contextual knowledge, the contextual representations of Teacher model can be well transferred to Student model. Compared with existing BERT compression methods, like MobileBERT/DistilBERT/TinyBERT, this CKD has the advantage of being directly applied on top of other pre-trained small BERT models without conducting time-consuming pre-training process.

The authors evaluate their approach on GLUE datasets and compare it to other state-of-the-art models.

The paper is well-written and organized, the experiments are thorough. However, I have several concerns:

* This proposed KD method is designed for the distillation on downstream tasks, so the whole distillation process should be conducted for each task, while the task-agnostic KD method, like MobileBERT, can be directly used with fine-tuning, please identify this fact in the introduction part. It would be more interesting, if experiments can be conducted during the pre-training stage and further evaluated.

* More experiments on challenging tasks like QA should be added.

* In the Table 1, the performance of TinyBERT on MNLI-mm is 82.6, while in an old version of tinybert paper, (https://arxiv.org/pdf/1909.10351v4.pdf), in the Table 10, the corresponding value is 83.2. And on the official GLUE benchmark the TinyBERT has comparable performance as the proposed CKD(w/DA).

* In the section 5.2, the MobileBERT is further improved by the proposed CKD with self-distillation, that is the MobileBERT is used as its own teacher on the downstream tasks. This comparison is not that fair, since MobileBERT can also be improved by other self-distillation method.

* In the section 5.3, “we observe that the BERTMINI trained with the CKD shows the higher average
score even though BERTMINI has fewer model parameters.” this comparison is unfair since BERTMINI has 6 layers and TinyBERT is a 4-layer model, and less number of model parameters does not always mean fast inference.

---

> ### Author Response · Authors · 2020-11-18
> **Response to Reviewer #1 (2/2)**
>
> **Q4: [The MobileBERT is further improved by the proposed CKD. ... This comparison is not that fair, since MobileBERT can also be improved by other self-distillation methods.]**
>
> - The goal of Section 5.2 is slightly different from that of Section 5.1, which compares several state-of-the-art distillation methods. Instead, in Section 5.2, we question whether we can further improve the fine-tuning of MobileBERT architecture, which currently shows the state-of-the-art performance on GLUE datasets, and our answer is affirmative with our CKD. The reason behind having a different goal here is the huge computational overhead of learning a teacher model in this setting (The original MobileBERT paper introduces a new architecture called IB-BERT to allow distillation matching for MobileBERT. Without this IB-BERT, it is impossible to compare diverse distillation methods with structural constraints, as we did in Section 5.1. Details of constraints for distillation objectives are described in Appendix A.)
>
> - However, we understand the concerns of the reviewer and in order to partially resolve them, we conducted additional experiments comparing our results in Section 5.2 against the basic KD using logit matching, which also has no architectural restrictions on teacher and student networks. Out of total 8 datasets, our approach showed better performance than logit KD on 5 datasets (Table 3 below shows the results on these datasets), but it was confirmed that there was no significant difference in the rest 3 datasets.
>
> - In fact, we think this is due to the fact that MobileBERT itself is state-of-the-art and has already saturated distillation performance to some extent, so it might be difficult to make a big difference across KD strategies. However, we want to note that our initial goal is only to check if we can boost the performance of MobileBERT, so there is room for additional tuning for our method. We will update as we get new results on this.
>
>
>
> Table 3. Results of knowledge distillation on Mobile BERT with possible objectives.
>
> | Method | CoLA (Mcc) | MNLI-m (Acc) | MNLI-mm (ACC) | RTE (Acc) | SST-2 (Acc) | QNLI (Acc) |
> |-|-|-|-|-|-|-|
> | BERT(BASE) | 60.4 | 84.8 | 84.6 | 70.4 | 94.0 | 91.8 |
> | (Teacher) |  |  |  |  |  |  |
> | MobileBERT | 54.0(±0.34) | 83.4(±0.09) | 83.8(±0.12) | 64.7(±0.41) | 92.1(±0.14) | 91.2(±0.12) |
> | MobileBERT |  |  |  |  |  |  |
> | (w/ Logit KD) | 54.2(±0.31) | 83.6(±0.04) | 84.1(±0.03) | 65.7(±0.38) | 92.1(±0.10) | 91.3(±0.06) |
> | (w/ CKD) | **54.8(±0.21)** | **84.1(±0.03)** | **84.3(±0.02)** | **67.1(±0.27)** | **92.3(±0.09)** | **91.4(±0.07)** |
>
> ---
>
> **Q5: [In the section 5.3, “we observe that the BERT (mini) trained with the CKD. … This comparison is unfair since BERT (mini) has 6 layers and TinyBERT is a 4-layer model, and less number of model parameters does not always mean fast inference.]**
>
> - Thank you for reviewing our statement carefully. For easier comparison with reference models such as TinyBERT, although not the main goal of this experiment, we also report FLOPs and corresponding speedup in Table 4 of the paper. The added columns are summarized in Table 4 below. At the same time, the sentence pointed out by the reviewer is toned down and changed as follows: **we observe that the BERT (mini) trained with the CKD achieves the higher average score while it has few model parameters with comparable but slightly higher FLOPs.**
>
>
>
> Table 4. The number of parameters and FLOPs of student models
> used in our experiments. These values are updated in the paper.
>
> |  Models | #Params | #FLOPs | Speedup |
> |-|:-:|:-:|:-:|
> | BERT (base) | 110.1M | 21754M | 1.00x |
> | BERT (6 layer) | 67.5M | 10878M | 2.00x |
> | BERT (small) | 29.1M | 3324M | 6.54x |
> | BERT (shallow) | 17.6M | 2419M | 8.99x |
> | TinyBERT | 14.5M | 1167M | 18.64x |
> | BERT (mini) | 12.5M | 1210M | 17.98x |
>
>
> **References**
> [1] Jiao, et.al. “TinyBERT: Distilling BERT for Natural Language Understanding”, EMNLP 2020, (https://arxiv.org/abs/1909.10351)
> [2] Sanh, et.al.“DistilBERT, a distilled version of BERT: smaller, faster, cheaper, and lighter”, NeurIPS Workshop 2019

---

> > ### Comment · AnonReviewer1 · 2020-11-24
> > **Most of my concerns have been addressed but improvement is marginal**
> >
> > Dear authors,
> >
> > Thanks for your efforts and extra experiments, the proposed two types of contextual knowledge is really interesting and I like this distribution-based  knowledge, and comprehensive experiments have been conducted, especially extra experiments were added in the new version.
> >
> > However, in comparison to some other distillation methods, e.g. the TinyBERT, this method is already not a strong baseline, and it is proposed about one year ago. As we know, it has been widely accepted that a thin-deep model tends to have better performance under the constraints of number of parameters, although the proposed BERT_{MINI} adopting a 6-layer architecture, the improvement over a 4-layer tinybert is still marginal. I understand that the authors may not have enough GPU servers to conduct expensive pre-training distillation, which may limit the further improvement of the proposed method.  Maybe, the authors can continue to run their algorithm on the released models of tinybert or some other more strong compressed models to verify the effectiveness of the proposed contextual knowledge.

---

> > > ### Author Response · Authors · 2020-11-24
> > > **Response to Reviewer #1 about additional comments**
> > >
> > > We sincerely thank you for mentioning that our approach is really interesting. We would like to clear up some misunderstandings to change your mind a little more positively.
> > >
> > > **Q1: However, in comparison to some other distillation methods, e.g. the TinyBERT, this method is already not a strong baseline, and it is proposed about one year ago**
> > >
> > > - We understand your concerns, but we emphasize that we are proposing the knowledge distillation objectives for Transformer compression. To the best of our knowledge, there are no studies that suggest new distillation objectives after TinyBERT and MobileBERT. Therefore, although it was proposed about one year ago, we do not agree that it is not a strong baseline for our specific purpose of comparison with Transformer.
> > >
> > > --------------------------------------------------------
> > >
> > > **Q2: As we know, it has been widely accepted that a thin-deep model tends to have better performance under the constraints of a number of parameters, although the proposed BERT_{MINI} adopting a 6-layer architecture, the improvement over a 4-layer tinybert is still marginal.**
> > >
> > >
> > > - Please note again that the purpose of experiments in experiment section 4.3, is **not to make comparisons against TinyBERT but to validate that our CKD works well in various architectures** without any constraints on distillation objectives. TinyBERT is simply shown as a reference for the model size, as explicitly mentioned in the paper.
> > >
> > > - Regarding the comparisons against other distillation objectives including TinyBERT, experiments on GLUE and SQuAD datasets are shown in Table 1 and Table 2 of the paper, respectively. As shown in Table 1 (Table 2 above), **our CKD shows significantly better performance than TinyBERT without overlapping standard errors** on all GLUE datasets except RTE and STS-B. Moreover, as shown in Table 1 above, our CKD has a **2.1 %p** improvement over TinyBERT on the SQuAD dataset. **We believe that these results support the claim that our distillation objectives outperform TinyBERT which is the current state-of-the-art distillation objectives.**
> > >
> > > ---
> > >
> > >
> > > **Q3: Maybe, the authors can continue to run their algorithm on the released models of tinybert or some other more strong compressed models to verify the effectiveness of the proposed contextual knowledge.**
> > >
> > > - Thank you for your constructive suggestions. To verify the effectiveness of the proposed contextual knowledge, as we mentioned in the paper, we conducted on the MobileBERT as shown in Table 3 of the paper (Table 3 above) and can boost the performance of the strong compressed models with our CKD.
> > > - In addition, to respond to reviewer 2’s comment, we experimented on the officially released TinyBERT and present in Table 2 of the response to reviewer 2. (We bring the Table below.) For this experiment, we use the same pre-trained TinyBERT (4 layers, 312 hidden states, 12 attention head) released in the official repository and perform task-specific distillation using each distillation objective. We note that, in the Table below, the performance of TinyBERT reported by the authors of [1] is also the result of conducting the task-specific distillation.  Although we do not use the data augmentation for TinyBERT and ours due to the limited time of rebuttal, our CKD significantly outperforms the TinyBERT, as shown in the Table below.
> > >
> > >
> > > | Method |$\hskip 0.4cm$ MRPC |$\hskip 0.25cm$ MNLI-m | MNLI-mm |$\hskip 0.4cm$ CoLA |
> > > |-|:-:|:-:|:-:|:-:|
> > > |  | (ACC) | (ACC) | (ACC) | (ACC) |
> > > | TinyBERT (Reported) | 82.4 | 80.5 | 81.0 | 29.8 |
> > > | CKD (all) | $\hskip 0.1cm$ **83.5(±0.18)** | $\hskip 0.1cm$ **81.1(±0.02)** | $\hskip 0.1cm$ **81.6(±0.01)** | $\hskip 0.1cm$ **37.9(±0.94)** |
> > >
> > > **Reference**
> > > [1] Jiao, et.al. “TinyBERT: Distilling BERT for Natural Language Understanding”, EMNLP 2020, (https://arxiv.org/abs/1909.10351)

---

> ### Author Response · Authors · 2020-11-18
> **Response to Reviewer #1 (1/2)**
>
> Thank you for the helpful suggestions. We address your concerns below:
>
> **Q1: [This proposed KD method is designed for the distillation on downstream tasks. …  Please identify this fact in the introduction part. It would be more interesting if experiments can be conducted during the pre-training stage and further evaluated.]**
>
> - Following the reviewer's suggestion, we clarify that our distillation process is focused on the task-specific distillation in the introduction part (See the introduction in the revised paper). We totally agree with the reviewer that using distillation for the pre-training stage makes our story much stronger. In fact, we naturally considered a task-agnostic approach from the beginning of this research, but it requires so many resources that we had to give up. While we could not conduct experiments with pre-training distillation only due to insufficient resources, we still believe that our contextual knowledge distillation can be directly applied for a task-agnostic approach and be one of the main components for other researchers in that field with huge resources.
>
> ---
>
> **Q2: [More experiments on challenging tasks like QA should be added.]**
>
> - Following the reviewer's suggestion, we additionally evaluate our method on SQuAD which is the standard QA dataset. As shown in Table 1, our CKD has a **2.1 %p** improvement over TinyBERT, which is the state-of-the-art knowledge distillation method. We included these results in the revision (See Table 2 of the paper and Table 1 below.)
>
>
> Table 1. Comparisons against state-of-the-art distillation methods on
> SQuAD 1.1v dataset (EM/F1 on dev set). For a fair comparison, all
> students are 6/768 BERT models, distilled by BERT base (12/768)
> teachers. The results of PKD and TinyBERT are as reported
> by Jiao et al. (2019) [1] and the result of DistilBERT is as reported
> by the author (Sanh et al., 2019) [2].
>
> | Method | #Params | $\hskip0.4cm$ #FLOPs | SQUAD 1.1 v |
> |-|:-:|:-:|:-:|
> |  |  | (Speed up) | EM $\hskip0.7cm$ F1 |
> | BERT (base)(Teacher) | 110 M | 21754M (x1.00) | 81.3 $\hskip0.5cm$ 88.6 |
> | PKD | 67.5M | 10878M (x2.00) | 77.1 $\hskip0.5cm$ 85.3 |
> | DistilBERT | 67.5M | 10878M (x2.00) | 79.1 $\hskip0.5cm$ 86.9 |
> | TinyBERT | 67.5M | 10878M (x2.00) | 79.7 $\hskip0.5cm$ 87.5 |
> | CKD (pairwise only) | 67.5M | 10878M (x2.00) | 81.3 $\hskip0.5cm$ 88.4 |
> | CKD (all) | 67.5M | 10878M (x2.00) | $\hskip0.1cm$ **81.8** $\hskip0.4cm$ **88.7** |
>
> ---
>
> **Q3: [In the Table 1, the performance of TinyBERT on MNLI-mm is 82.6, while in an old version of tinybert paper. … On the official GLUE benchmark the TinyBERT has comparable performance as the proposed CKD(w/DA).]**
>
>
> - Thank you for letting us know that. We correct the misreported performance of MNLI-mm to 83.2 and the performance of RTE, which was updated in the new version of TinyBERT after the ICLR submission deadline. However, ours shows 84.0 on the MNLI-mm dataset, and it still outperforms TinyBERT on 7 out of 8 datasets (previously all datasets, but the result on RTE in the new version of TinyBERT improves a lot) as well as the final average. In order to have more convincing results, we additionally present the **standard errors** of our methods on GLUE **devset** in Table 2 below. The result shows that our CKD shows significantly better performance than TinyBERT without overlapping standard errors. Moreover, as shown in Table 1 above, our CKD has a **2.1 %p** improvement over TinyBERT on the SQuAD dataset.
>
>
>
> Table 2. Comparisons against state-of-the-art distillation methods on the development set. We report the means of performances
> which are averaged over 5 runs and standard errors. * indicates that value is changed following a revised paper [1]
> after ICLR submission deadline.
>
> | Method | CoLA | MNLI-(m/mm) | SST-2 | QNLI | MRPC | QQP | RTE | STS-B |
> |-|-|-|-|-|-|-|-|-|
> | BERT(base) | 60.4 | 84.8/84.6 | 94.0 | 91.8 | 90.3 | 91.4 | 70.4 | 89.5 |
> | (Teacher) |  |  |  |  |  |  |  |  |
> | DistilBERT | 42.5 | 81.6/81.1 | 92.7 | 85.5 | 88.3 | 90.6 | 60.0 | 85.0 |
> | PD | - | 82.5/83.4 | 91.1 | 89.4 | 89.4 | 90.7 | 66.7 | - |
> | TinyBERT(w/ DA) | 54.0 | 84.5/84.5 | 93.0 | 91.1 | 90.6 | 91.1 | **73.4**\* | **89.6** |
> | CKD | 52.8(±0.89) | 83.9/84.4(±0.10) | 93.3(±0.13) | 90.5(±0.11) | 89.6(±0.21) | 90.9(±0.11) | 67.3(±0.97) | 89.0(±0.08) |
> | CKD(w/ DA) | **57.9(±0.46)** | **84.8/85.0(±0.04)** | **93.8(±0.08)** | **91.7(±0.06)** | **90.8(±0.11)** | **91.6(±0.03)** | 70.1(±0.41) | **89.6(±0.07)** |

---

### Official Review · AnonReviewer2 · 2020-10-27
**Doesn't seem too impactful**

**Rating:** 5
**Confidence:** 3

**Review:**

This paper presents a new knowledge distillation (KD) method for distilling BERT. This area is pretty active given that deploying BERT based models is of keen interest to many industrial applications.

This paper proposes distillation via modeling "Word Relation and Layer Transforming Relation" which essentially aims to "capture the knowledge of relationships between word representations and LTR defines how each word representation changes
as it passes through the network layers". Not to mention *pairwise* and triple-wise relations are being modeled

My biggest question in the paper (which the doesn't paper address) is that this is bound to be expensive. Yet there is hardly any mention of training time (or time needed to cache these values from the teacher).

This seems even worse when there is not much gain over existing baselines and can be attributed to simply noise/variance.

Overall, I don't think this method will be impactful at all and it is probably not worth having over existing approaches. It is far too complex. Experiments on the GLUE benchmark alone is also not convincing.

The authors can try other tasks and perhaps SuperGLUE to make their experiments more convincing.

I think a runtime analysis could help to make the paper stronger to understand the differences. But I would like to make it clear that I want a honest, runtime analysis of how long this distillation would take or the practical considerations. How much more expensive is it to align and compute these values during training of the student. Please make this clear.

I would also like to see a runtime analysis of the baselines as well.

---

> ### Author Response · Authors · 2020-11-18
> **Response to Reviewer #2 (2/2)**
>
> **Q3: [Gains can be attributed to simply noise/variance.]**
>
> - Please note that the development set results of Table 1 in our paper are based on **5 runs**to reduce the experimental noise. In response to your concerns, we additionally report the **standard errors**of our experiments in Table 2 above and Table 4 below. As shown in Table 4, for all datasets except for RTE and STS-B, our CKD shows significantly better performance than TinyBERT without overlapping standard errors.
>
>
> Table 4. Comparisons against state-of-the-art distillation methods on the development set. We report the means of performances
> which are averaged over 5 runs and standard errors. * indicates that value is changed following a new version of
> TinyBERT[2] which was revised after ICLR submission deadline.
>
> | Method | CoLA | MNLI-(m/mm) | SST-2 | QNLI | MRPC | QQP | RTE | STS-B |
> |-|-|-|-|-|-|-|-|-|
> | BERT(base) | 60.4 | 84.8/84.6 | 94.0 | 91.8 | 90.3 | 91.4 | 70.4 | 89.5 |
> | (Teacher) |  |  |  |  |  |  |  |  |
> | DistilBERT | 42.5 | 81.6/81.1 | 92.7 | 85.5 | 88.3 | 90.6 | 60.0 | 85.0 |
> | PD | - | 82.5/83.4 | 91.1 | 89.4 | 89.4 | 90.7 | 66.7 | - |
> | TinyBERT(w/ DA) | 54.0 | 84.5/84.5 | 93.0 | 91.1 | 90.6 | 91.1 | **73.4**\* | **89.6** |
> | CKD | 52.8(±0.89) | 83.9/84.4(±0.10) | 93.3(±0.13) | 90.5(±0.11) | 89.6(±0.21) | 90.9(±0.11) | 67.3(±0.97) | 89.0(±0.08) |
> | CKD(w/ DA) | **57.9(±0.46)** | **84.8/85.0(±0.04)** | **93.8(±0.08)** | **91.7(±0.06)** | **90.8(±0.11)** | **91.6(±0.03)** | 70.1(±0.41) | **89.6(±0.07)** |
>
> ---
>
> **Q4: [Experiment on the GLUE benchmark alone is also not convincing.]**
>
> - Following the reviewer's suggestion, we additionally evaluate our method on SQuAD which is the standard QA dataset. As shown in Table 5, our CKD has a **2.1 %p**improvement over TinyBERT, which one is the state-of-the-art knowledge distillation method. We updated these results to the paper. (See Table 2 of the paper)  This result also supports the claim that the improvement of performance in our CKD is not attributed to noise.
>
>
> Table 5. Comparisons against state-of-the-art distillation methods on
> SQuAD 1.1v dataset (EM/F1 on dev set). For a fair comparison, all
> students are 6/768 BERT models, distilled by BERT base (12/768)
> teachers. The results of PKD and TinyBERT are as reported  by
> Jiao et al. (2019) [2] and the result of DistilBERT is as reported
> by the author (Sanh et al., 2019) [3].
>
> | Method | #Params | $\hskip0.4cm$ #FLOPs | SQUAD 1.1 v |
> |-|:-:|:-:|:-:|
> |  |  | (Speed up) | EM $\hskip0.7cm$ F1 |
> | BERT (base)(Teacher) | 110 M | 21754M (x1.00) | 81.3 $\hskip0.5cm$ 88.6 |
> | PKD | 67.5M | 10878M (x2.00) | 77.1 $\hskip0.5cm$ 85.3 |
> | DistilBERT | 67.5M | 10878M (x2.00) | 79.1 $\hskip0.5cm$ 86.9 |
> | TinyBERT | 67.5M | 10878M (x2.00) | 79.7 $\hskip0.5cm$ 87.5 |
> | CKD (pairwise only) | 67.5M | 10878M (x2.00) | 81.3 $\hskip0.5cm$ 88.4 |
> | CKD (all) | 67.5M | 10878M (x2.00) | $\hskip0.1cm$ **81.8** $\hskip0.4cm$ **88.7** |
>
> **References**
> [1] Sun, et.al. Patient Knowledge Distillation for BERT Model Compression, EMNLP 2019
> [2] Jiao, et.al  TinyBERT : Distilling BERT for Natural Language Understanding, EMNLP 2020
> [3] Sanh, et.al.“DistilBERT, a distilled version of BERT: smaller, faster, cheaper, and lighter”, NeurIPS Workshop 2019

---

> > ### Comment · AnonReviewer2 · 2020-11-23
> > **Thanks for the extra experiments**
> >
> > Thanks for performing the extra experiments. I have increased my score to 5. The full method here is still pretty expensive (50% more). Pairwise only outperforms tinybert while maintaining training time but the gain isn't that significant anymore. In terms of compute-performace trade-off, this is not looking so good. Nevertheless, the new experiments are helpful and therefore I raised my score to 5.

---

> > > ### Author Response · Authors · 2020-11-23
> > > **Thanks for the additional comments**
> > >
> > > We sincerely thank you for your time and effort in additional comments.
> > >
> > > We understand your concerns on training time, but please note again that **the ultimate purpose of the model compression including baselines and our paper is to reduce the computation and memory costs when the network is deployed at inference time after training.**
> > >
> > > In this line of spirit, several works employed meta-learning [1] or adversarial training [2, 3], which require a lot of additional training times to improve the performance of the student network. In addition, very recently, Wang et al. [4] improve the performance of student networks on multi-lingual sequence labeling tasks through the sequence-level probability distillation which predicts the top-k best label sequences. According to the authors of [4], their algorithm requires 0.5x additional training times as shown in the below Table. (We cite the table from the paper [4]).
> > >
> > > |  | Training Time (hours) |
> > > |:-:|:-:|
> > > | Baseline | 11 |
> > > | TOP-WK [4] | 18 |
> > > | POSTERIOR [4] | 16 |
> > >
> > > Regarding the performance of pairwise only CKD, as shown in Table 2 above, please note that pairwise only CKD outperforms the TinyBERT without overlapping standard errors on all datasets. Moreover, on the CoLA (Table 2 above) and SQuAD (Table 5 above) datasets, our CKD using only pairwise has **6.5%p**and **1.6%p**improvement over TinyBERT, respectively. **We believe that even pairwise results are not marginal improvements.**
> > >
> > >
> > > In short, the efficiency-performance trade-off should be considered in terms of **test/inference time when deployed**.
> > >
> > > If your concern is only training times of our method, please consider it again and let us know if there is anything else that you want to discuss further.
> > >
> > > Thank you again for your time and please keep safe!
> > >
> > >
> > > **Reference**
> > > [1] Jang et.al., Learning what and where to transfer, ICML 2019
> > > [2] Heo et.al., Knowledge Distillation with Adversarial Samples Supporting Decision Boundary, AAAI 2019
> > > [3] Gao et.al., An Adversarial Feature Distillation Method for Audio Classification, IEEE Access 2019
> > > [4] Wang et.al., Structure-Level Knowledge Distillation For Multilingual Sequence Labeling, ACL 2020

---

> ### Author Response · Authors · 2020-11-18
> **Response to Reviewer #2 (1/2)**
>
> Thank you for the review. We address your comments below:
>
> Please note that the purpose of model compressions, such as sparsification or quantization, as well as distillation, is to reduce the computation and memory costs when the network is deployed at **inference time**(possibly under the resource-constrained environments). Therefore, the majority of papers in this field (including all baselines considered in our paper) focus only on the cost and the performance at test time. We do not agree with the reviewer’s claim that our method is “not impactful at all” or “not worth having” due to the expensive computations at training time.
>
> Moreover, even the belief that our method is very expensive in training time is a misunderstanding that might come from a seemingly rather complicated formula. In fact, it is only in the form of a simple summation over additional pairwise (and triplewise) terms among few entries, so it does not hurt the training process as badly as the reviewer concerned. In order to clear the reviewer’s misunderstanding, we have conducted additional experiments as the reviewer requested, so please take a closer look and ask additional questions for any uncertain parts.
>
> ---
>
> **Q1: [My biggest question in the paper is that this is bound to be expensive. Yet there is hardly any mention of training time]**
>
> - Following the reviewer's suggestion, we report the training time per iteration of our method and baselines in Table 1 below. Fixing the hardware (Titan RTX) and hyperparameters for fair comparisons, we observe that our distillation method takes almost 1.5x times compared to baselines. Here we can confirm that calculating the triplet relation consumes most of the additional times. (See CKD (all) and CKD (pairwise only))
>
> - Since someone might think that the increase of 0.5x training time is too big, we additionally evaluate the performance of our CKD using **only pairwise**relations over TinyBERT, which one is the current state-of-the-art methods with similar training time. For a fair comparison, we use the same pre-trained TinyBERT (4 layers, 312 hidden states, 12 attention head) released in the official repository, and perform task-specific distillation using each distillation objective. As shown in Table 2 below, CKD using only **pair-wise relations can outperform TinyBERT on GLUE datasets**while the triple-wise term is also required to further boost the performance.
>
>
> Table 1. Comparison of baselines and CKD
> about the training time.
>
> | Method | Training Time (ms/iter) |
> |-|:-:|
> | KD (Logit KD) | 206.3 |
> | DistilBERT | 241.7 |
> | PKD | 235.5 |
> | TinyBERT | 264.2 |
> | CKD (all) | 334.6 |
> | CKD (pairwise only) | 243.1 |
>
>
> Table 2. Comparison of TinyBERT and CKD on the subset of GLUE with training time. The performance of
> TinyBERT is cited as reported by the authors.
>
> | Method | Training Time |$\hskip 0.4cm$ MRPC |$\hskip 0.25cm$ MNLI-m | MNLI-mm |$\hskip 0.4cm$ CoLA |
> |-|:-:|:-:|:-:|:-:|:-:|
> |  | (ms/iter) | (ACC) | (ACC) | (ACC) | (ACC) |
> | TinyBERT (Reported) | 264.2 | 82.4 | 80.5 | 81.0 | 29.8 |
> | CKD (pairwise only) | 243.1 | 82.7(±0.23) | 80.9(±0.03) | 81.3(±0.02) | 36.3(±0.76) |
> | CKD (all) | 334.6 | $\hskip 0.1cm$ **83.5(±0.18)** | $\hskip 0.1cm$ **81.1(±0.02)** | $\hskip 0.1cm$ **81.6(±0.01)** | $\hskip 0.1cm$ **37.9(±0.94)** |
>
> ---
>
> **Q2: [I think a runtime analysis could help to make the paper stronger to understand the differences. … How much more expensive is it to align and compute these values during training of the student.]**
> - As shown in Table 1 above, there is an increase of around 0.5 times the total training time by using our distillation method. Here, Table 3 reports the time spent in each component in the training. While computing the triple-wise relation in a forward pass requires x5.2 times compared to baselines, the **main bottleneck in the training is the forward and backward passes of the networks.** Again, as we mentioned above, the ultimate purpose of network compression is to reduce the computation and memory costs at inference time after training.
>
> - Regarding the alignment, we basically use the same strategy with [1,2]. Moreover, the alignment process does not affect much the overall training time as shown in Table 3.
>
>
> Table 3. Required time to process each
> operation in our method and baselines
>
> | Process | Required Time |
> |-|:-:|
> |  | (ms/iter) |
> | Teacher Forward | 16.3 ms |
> | Student Forward | 9.3 ms |
> | Alignment | 0.29 ms |
> |$\textit{Loss function}$ |  |
> | Logit KD | 0.89 ms |
> | PKD Loss | 1.10 ms |
> | TinyBERT Loss | 1.82 ms |
> | CKD (Pair-wise) | 1.68 ms |
> | CKD (Triple-wise) | 9.42 ms |

---

### Official Review · AnonReviewer4 · 2020-10-28
**CKD Review**

**Rating:** 6
**Confidence:** 4

**Review:**

The paper proposed a contextual knowledge distillation approach by leveraging two types of contextual knowledge: word relations and layer transforming relation. Recent advancement in this area emphasizes the promising effect of this area in language modeling.

The paper is well-written and well-structured. The experiment section shows a complete set of experiments including the baselines, benchmark and ablation study. The results are relatively incremental in comparison with TinyBert. Considering that the improvement has been relatively incremental, it would be helpful to compare the models with respect to FLOPs and speedup.
Novelty: It seems that the notion of structural knowledge distillation have been used previously by Wang et al [1]. It would be great if the authors clarify about their contribution. Also,    the related work section can be enriched by new publications such as PoWER-BERT [2], FastBert [3] and TextBrewer[4]
1)	Structure-Level Knowledge Distillation For Multilingual Sequence Labeling, ACL 2020
2)	PoWER-BERT: Accelerating BERT Inference via Progressive Word-vector Elimination , ICML 2020
3)	FastBERT: a Self-distilling BERT with Adaptive Inference Time , acl 2020
4)	TextBrewer: An Open-Source Knowledge Distillation Toolkit for Natural Language Processing

---

> ### Author Response · Authors · 2020-11-18
> **Response to Reviewer #4**
>
> Thank you for the review. We address your concerns below:
>
> **Q1: [The results are relatively incremental in comparison with TinyBert.]**
> - In order to have more convincing results, we additionally report the **standard errors**of our method on the development set of GLUE. As shown in Table 1 below, for all datasets except for the RTE and STS-B, our CKD shows a significantly better performance than TinyBERT without overlapping the standard errors. Therefore, we believe that our results are not a marginal improvement.
> - Moreover, from the additional experiment on the SQuAD which is a question and answering dataset, we show that our CKD has a **2.1 %p**improvement over TinyBERT on the SQuAD dataset. We add this experiment in Table 2 of the revised paper and Table 2 below.
>
>
> Table 1. We report the means of performances which are averaged over 5 runs and standard errors on the
> development set of GLUE. \* indicates that value is changed following a revised paper [1] after our submission.
>
> | Method | CoLA | MNLI-(m/mm) | SST-2 | QNLI | MRPC | QQP | RTE | STS-B |
> |-|-|-|-|-|-|-|-|-|
> | BERT(base) | 60.4 | 84.8/84.6 | 94.0 | 91.8 | 90.3 | 91.4 | 70.4 | 89.5 |
> | (Teacher) |  |  |  |  |  |  |  |  |
> | DistilBERT | 42.5 | 81.6/81.1 | 92.7 | 85.5 | 88.3 | 90.6 | 60.0 | 85.0 |
> | PD | - | 82.5/83.4 | 91.1 | 89.4 | 89.4 | 90.7 | 66.7 | - |
> | TinyBERT(w/ DA) | 54.0 | 84.5/84.5 | 93.0 | 91.1 | 90.6 | 91.1 | **73.4**\* | **89.6** |
> | CKD | 52.8(±0.89) | 83.9/84.4(±0.10) | 93.3(±0.13) | 90.5(±0.11) | 89.6(±0.21) | 90.9(±0.11) | 67.3(±0.97) | 89.0(±0.08) |
> | CKD(w/ DA) | **57.9(±0.46)** | **84.8/85.0(±0.04)** | **93.8(±0.08)** | **91.7(±0.06)** | **90.8(±0.11)** | **91.6(±0.03)** | 70.1(±0.41) | **89.6(±0.07)** |
>
>
> Table 2. Results on SQuAD 1.1v dataset (EM/F1 on dev set). For a
> fair comparison, all students are 6/768 BERT models, distilled by
> BERT base (12/768) teachers. The results of PKD and TinyBERT are
> as reported by Jiao et al. (2019) [1] and the result of DistilBERT is
> as reported by the authors (Sanh et al., 2019) [2].
>
> | Method | #Params | $\hskip0.4cm$ #FLOPs | SQUAD 1.1 v |
> |-|:-:|:-:|:-:|
> |  |  | (Speed up) | EM $\hskip0.7cm$ F1 |
> | BERT (base)(Teacher) | 110 M | 21754M (x1.00) | 81.3 $\hskip0.5cm$ 88.6 |
> | PKD | 67.5M | 10878M (x2.00) | 77.1 $\hskip0.5cm$ 85.3 |
> | DistilBERT | 67.5M | 10878M (x2.00) | 79.1 $\hskip0.5cm$ 86.9 |
> | TinyBERT | 67.5M | 10878M (x2.00) | 79.7 $\hskip0.5cm$ 87.5 |
> | CKD (pairwise only) | 67.5M | 10878M (x2.00) | 81.3 $\hskip0.5cm$ 88.4 |
> | CKD (all) | 67.5M | 10878M (x2.00) | $\hskip0.1cm$ **81.8** $\hskip0.4cm$ **88.7** |
>
> ---
>
> **Q2: [Considering that the improvement has been relatively incremental, it would be helpful to compare the models with respect to FLOPs and speedup.]**
>
> - Please note that we follow the **standard setup**for baselines, which use the 6-layer BERT as the student network for a fair comparison. Therefore, in Table 1 of the paper, **the student models used in all baselines and our methods have the same number of FLOPs (10878M) and speed up (2.00x).** In response to the reviewer's comment, we also report FLOPs and corresponding speedup in Table 4 of the paper which shows the effect of our CKD on various sizes of network architecture, as summarized in Table 3 below.
>
>
> Table 3. The number of parameters and FLOPs
> of student models used in our experiments.
>
> |  Models | #Params | #FLOPs | Speedup |
> |-|:-:|:-:|:-:|
> | BERT (base) | 110.1M | 21754M | 1.00x |
> | BERT (6 layer) | 67.5M | 10878M | 2.00x |
> | BERT (small) | 29.1M | 3324M | 6.54x |
> | BERT (shallow) | 17.6M | 2419M | 8.99x |
> | TinyBERT | 14.5M | 1167M | 18.64x |
> | BERT (mini) | 12.5M | 1210M | 17.98x |
> ---
> **Q3: [Novelty: It seems that the notion of structural knowledge distillation have been used previously by Wang et al [3]. It would be great if the authors clarify about their contribution.]**
>
> - Thank you for letting us know that. Wang, et.al. [3] proposes the method that distills the predictive distribution of sequence-level for the multilingual sequence labeling task. In our understanding, the wording of “structure”  in [3] is used to describe the sentence-level prediction probability. Therefore, our proposed distillation objectives which design the relationship in the word representations are totally different from Wang, et.al [3]. We added this to the related work in the revision.
>
> ---
> **Q4: [Also, the related work section can be enriched by new publications such as PoWER-BERT, FastBert and TextBrewer]**
>
> - Thank you for suggesting the additional related work. We cite them and clarify the relevance of our paper.
>
> **References**
> [1]Jiao, et.al. “TinyBERT: Distilling BERT for Natural Language Understanding”, EMNLP 2020
> [2] Sanh, et.al.“DistilBERT, a distilled version of BERT: smaller, faster, cheaper, and lighter”, NeurIPS Workshop 2019
> [3] Wang, et.al. Structure-Level Knowledge Distillation For Multilingual Sequence Labeling, ACL 2020

---

### Author Response · Authors · 2020-11-20
**Update Summary in the revision**

Thank you to all reviewers for their constructive and helpful feedback. Based on their comments, we revised the paper by making the following changes. (The changed part is highlighted in **red**.) :

**Major updates**
- We have included the content in the introduction part that we focus on the task-specific distillation with the advantage of not conducting a time-consuming pre-training process, as suggested by Reviewer 1. (R1)

- We have included the additional relevant work (Wang et al., 2020, Goyal et al., 2020; Liu et al., 2020; Hou et al., 2020) to clarify our contribution, as suggested by Reviewer 4. (R4)

- We have included an additional experiment on the Stanford Question Answering Dataset (SQuAD) in Table 2 of the paper, as suggested by Reviewers 1 and 2. (R1, R2)

- We have included FLOPs and speed up for student models in Table 2 and 4, as suggested by Reviewers 1, 3 and 4 (R1, R3, R4).

- The sentence comparing TinyBERT and our method in section 4.3 has been toned down and changed, as suggested by Reviewer 1. (R1)

**Minor updates**
- We have corrected typos and errors.

- We have corrected the performance of TinyBERT on MNLI-mm and RTE which was revised after ICLR submission deadline.


We believe that our paper gets much stronger and clearer with this revision, thanks to the reviewers for constructive suggestions.

---

### Author Response · Authors · 2020-11-23
**The end of the discussion phase approaching**

Dear Reviewers and Area Chair,

Could you please go over our responses and the revision since we can have interactions with you only by this Tuesday (24th)? We have responded to your comments and faithfully reflected them in the revision, and provided additional experimental results. We sincerely thank you for your time and efforts in reviewing our paper, and your insightful and constructive comments.

Thanks, Authors

---

### Decision · Program_Chairs · 2021-01-07
**Final Decision**

**Decision:**

Reject

**Comment:**

This paper proposes a new method to perform knowledge distillation (KD) for transformer compression, where two types of contextual knowledge, namely, word relations and layer-transforming relations, are considered for KD. Both pair-wise and triple-wise relations are modeled.

This paper receives two weak reject and two weak accept recommendations. On one hand, the reviewers appreciate that the authors have added more results into the paper to solve their concerns. On the other hand, several concerns still exist. (i) With regards to the compute-performance trade-off, the gains of the method does not seem too great. One reviewer feels that the authors tried to downplay the cost of their method too much. Though we care more about the inference time, the development time in practice should also not be underestimated. (ii) Compared with TinyBERT, the performance gain looks marginal on the GLUE benchmark (Table 1). (iii) It will make the paper more convincing if pre-training experiments can be performed.

Overall, after reading the paper, the AC thinks that the novelty of the proposed method is somewhat limited. The AC is also hesitant about whether modeling word relations and layer-transforming relations simultaneously are needed. The choices for ablation study are also not totally clear.

For example, in Figure 2, it is not clear why the authors choose SST-2 to plot the figure; in Table 5, it is unclear why SST-2, MRPC and QNLI are selected, but not others. When looking at Table 5, it is not totally convincing it is needed to model both WR and LTR, or it is needed to introduce both pair-wise and triple-wise relations. More careful ablation studies are needed. It also remains unclear what kind of word relations or layer-transforming relations are learned.

In summary, this is a borderline paper, and the rebuttal unfortunately did not fully address the reviewers' main concerns. On balance, the AC regrets that the paper cannot be recommended for acceptance at this time. The authors are encouraged to consider the reviewers' comments when revising the paper for submission elsewhere.